# Evaluation of the Effects of Repetitive Anaesthesia Administration on the Brain Tissues and Cognitive Functions of Rats with Experimental Alzheimer’s Disease [note 1]

**DOI:** 10.3390/medicina60081266

**Published:** 2024-08-05

**Authors:** Nuray Camgoz Eryilmaz, Mustafa Arslan, Aysegul Kucuk, Ayca Tas Tuna, Sevin Guney, Gulnur Take Kaplanoglu, Mustafa Kavutcu

**Affiliations:** 1Department of Anesthesiology and Reanimation, Gazi University Faculty of Medicine, 06500 Ankara, Türkiye; camgoznuray@gmail.com (N.C.E.); mustarslan@gmail.com (M.A.); 2Department of Physiology, Kutahya Health Sciences University Faculty of Medicine, 43020 Kutahya, Türkiye; 3Department of Anesthesiology and Reanimation, Sakarya University Faculty of Medicine, 54050 Sakarya, Türkiye; aycatas@yahoo.com; 4Department of Physiology, Gazi University Faculty of Medicine, 06500 Ankara, Türkiye; sguney@gazi.edu.tr; 5Department of Histology and Embryology, Gazi University Faculty of Medicine, 06500 Ankara, Türkiye; gulnurtake@gmail.com; 6Department of Medical Biochemistry, Gazi University Faculty of Medicine, 06500 Ankara, Türkiye; kavutcu@gazi.edu.tr

**Keywords:** Alzheimer, ketamine, propofol, radial arm maze

## Abstract

*Introduction:* We evaluated the effects of repeated ketamine, propofol, and ketamine + propofol administration on cognitive functions and brain tissue of elderly rat models with streptozotocin-induced Alzheimer’s disease. *Materials and Methods:* Thirty elderly male Wistar Albino rats were divided into five groups: control (Group C), Alzheimer’s (Group A), Alzheimer’s + ketamine (Group AK), Alzheimer’s + propofol (Group AP), and Alzheimer’s + propofol + ketamine (Group APK). Alzheimer’s disease was induced in Groups A, AK, AP, and APK via intracerebroventricular streptozotocin. Four weeks after surgery, ketamine, propofol, and ketamine + propofol were administered intraperitoneally for 3 days to Groups AK, AP, and APK, respectively. The radial arm maze test (RAMT) was performed in the initial, 1st, 2nd, 3rd, and 4th weeks after surgery and daily following anaesthesia. Blood and brain tissue samples were obtained. *Results:* The RAMT results of Groups A, AK, AP, and APK decreased compared to Group C 2 weeks after Alzheimer’s disease onset. Compared to Group A, the RAMT results increased in Groups AK and APK after the first anaesthesia, and in Group AP after the second anaesthesia. Brain tissue paraoxonase-1 (PON-1) and catalase (CAT) activities were low, and the thiobarbituric acid reactive substance (TBARS) level was high in Group A compared to Group C. TBARS levels of Groups AP and APK were lower than Group A, while CAT activity was higher. PON-1 activity was higher in Groups AK, AP, and APK than in Group A. Histopathological changes decreased in Groups AP and AK. A decrease in p53 was found in Group C compared to Group A. Ketamine and propofol were found to be effective at Bcl-2 immunoexpression, but a decrease in Caspase-3 was observed in Group APK. GFAP immunoexpression increased in Group A compared to Group C and in Group AP compared to Group AK. *Conclusions:* Repetitive anaesthesia application was found to positively affect cognitive functions. This was supported by histopathological and biochemical markers.

## 1. Introduction

Alzheimer’s disease (AD) is the most frequent form of dementia and the most common neurodegenerative illness in the aged population. Its characteristics are synapses loss, aberrant amyloid-beta peptide (Aβ) processing by the Aβ precursor protein (APP), and tau hyperphosphorylation [1]. Inflammatory processes and many other contributing factors are mentioned in the aetiology of AD. Aβ accumulation, caspase activation, and apoptosis have all been linked to AD neuropathogenesis. However, the role of anesthetic agents is not entirely understood. Many studies have implied a link between surgery, general anaesthesia (GA), and the development of AD symptoms [2,3,4,5,6,7]. Exposure to anesthetic drugs during surgery has been proven in animal studies and some human clinical trials to produce postoperative cognitive decline (POCD). Nevertheless, it is unclear whether this transitory decrease can lead to dementia over time.

In vitro and animal models can provide insights into the mechanisms through which GA influences AD neuropathology. It has been suggested that volatile anesthetics may cause an increase in Aβ production and Aβ-related neurotoxicity [8,9,10]. Studies have also shown that isoflurane and propofol can enhance tau hyperphosphorylation [11]. A few human studies have shown that tau concentrations in the cerebrospinal fluid significantly increase with cognitive decline after general anaesthesia, while Aβ levels remain unchanged [12,13]. Interactions between anaesthesia and acetylcholine receptors, such as muscarinic and nicotinic receptors, may also contribute to cognitive impairment [14]. As muscarinic receptors are implicated in the synthesis of Aβ, it is possible that anesthetics indirectly impact Aβ processing through the cholinergic system [15]. These findings indicate the inhibition of central cholinergic transmission, which is compromised in the elderly population. GA has been postulated to hasten the course of AD in this manner. Furthermore, N-methyl-D-aspartate receptor (NMDAR) activation has been shown to play a role in AD-related synaptic dysfunction. Thus, the NMDAR is believed to be involved in memory function and a cause of AD progression.

Some NMDAR antagonists such as memantine and ketamine have neuroprotective effects. However, it is still debated whether the concentration-dependent effects of NMDAR antagonists can be described in terms of synaptic physiology [1]. Both ketamine and memantine were shown to regulate long-term potentiation and depression and promote cognitive functions, with therapeutic effects on major depression and AD [16,17]. It has been reported that gamma-aminobutyric acid (GABA) reduces in AD patients, depending on their age, and that this decrease may be related to reduced protein levels and mRNA in GABA receptor subunits [18]. Propofol (2,6-disopropylphenol) is a GABA receptor agonist that may inhibit caspase-3 activation and Aβ oligomerization caused by isoflurane anaesthesia [19]. It has been shown to assist in the development of cognitive function in humans [20], and its effects on learning and memory have been investigated in many studies. However, these remain controversial [21,22,23,24,25,26].

Alzheimer’s disease is a condition that can be encountered in daily anesthesia practice in the elderly population. Ketamine and propofol are commonly used intravenous anesthetic agents in all age groups today. As postoperative cognitive dysfunction is more likely to be observed in Alzheimer’s patients, it is important to investigate the effects of ketamine and propofol [4]. The literature in this area lacks studies on how the combination of ketamine and propofol affect animals with AD. Therefore, the present study was conducted to evaluate the effects of repetitive anaesthesia, ketamine, and propofol administration on the cognitive functions of rats with AD.

## 2. Materials and Methods

This study was conducted with the ethical consent of the Gazi University Animal Experiments Ethics Committee (GÜET-19-002). At all stages of the investigation, the accepted standards of the Guide for the Care and Use of Laboratory Animals were followed. All study protocols were conducted in accordance with the UK Animals (Scientific protocols) Act 1986, the ARRIVE guidelines, and the EU Directive on animal testing. In this study, 30 elderly (22 months old) male Wistar Albino rats weighing 325–375 g were used. Two authors (N.C.Y and M.A) were aware of the grouping at the time of allocation, conduct of the trial, and outcome assessment. Histological and biochemical evaluation and data analysis were conducted by investigators blinded to this study. 

### 2.1. Groups and Protocol

A total of 30 rats were used. The rats were housed in the laboratory at 20–21 °C for 12 h in daylight and 12 h in darkness. Free access to food was allowed. For the experiment, fasting was provided for 2 h before the application of anesthesia. Each rat was numbered, and they were divided into five groups using the sealed-envelope method, with six rats in each: control group (Group C), AD group (Group A), AD + ketamine group (Group AK), AD + propofol group (Group AP), and AD + propofol + ketamine group (Group APK). All rats were given 100 mg/kg of ketamine intraperitoneally (i.p.), and a stereotactic head was inserted beneath the dura of each rat via a burr hole from the midline. According to the stereotaxic atlas of rat brain, the coordinates for the intracerebroventricular (icv) region were determined as 0.8 mm, 1.5 mm, and 3.8 mm in the antero-posterior, lateral, and dorsoventral directions, respectively [27]. The precision of the coordinates was confirmed through the icv application of methylene blue in preliminary experiments. Subsequently, a small incision was made in the anteroposterior direction of the skin over the skulls of the animals placed in the stereotaxic apparatus. With reference to the intersection point of bregma and sagittal suture on the skull, a hole was drilled with a stereotaxic drill at the point corresponding to the abovementioned coordinates. Streptozotocin (STZ) was then injected using a Hamilton injector by advancing 3.8 mm in the dorsoventral direction. Then, the incision was closed with three sutures. In line with previous studies, experimental AD was induced by intracerebroventricularly administering 3 mg/kg (10 μL) of streptozotocin (Sigma Chemical, St. Louis, MO, USA) to Groups A, AK, AP, and APK [28,29]. The STZ groups were administered icv injections of STZ dissolved in citrate buffer (pH 4.4), with the STZ concentration calculated to deliver 5 μL of solution per injection site. The same amount of citrate buffer was injected intracerebroventricularly into the rats in Group C.

The RAMT, which was designed by Olton and Samuelson, has been commonly used by researchers for measuring memory and spatial learning in rodents. The original design for the radial arm maze test (RAMT) includes a central platform 34 cm in width, with eight arms of equal length emerging from it. A food site that cannot be seen from the central area is situated at the end of each arm; food is placed at the distal end on four arms, while the other four arms are devoid of food. It is important to note that an animal must have access to food from all four areas of the maze to complete the RAMT [29,30,31]. In the present study, the animals were deprived of food for 2 h prior to the experiment, following which pellet food was placed at the ends of each arm of the maze. The rats were educated in the radial arm maze (RAM) for 300 s during the first 3 days of the experiment. RAMT was then performed with all groups in the initial, 1st, 2nd, 3rd, and 4th weeks after surgery and daily following anaesthesia; the input–output numbers of the rats in the arms were recorded. Four weeks after surgery, the anaesthetic agents, including 100 mg/kg of ketamine (ketalar 5%; Parke–Davis; Pfizer, Inc, New York, NY, USA) and 100 mg/kg of propofol (propofol 2%, Fresenius Kabi, Bad Homburg, Germany), were applied i.p. for 3 days to Groups AK, AP, and APK. The propofol and ketamine doses were guided by Yu DJ et al.’s and Arras M et al.’s recommendations, respectively, but the doses actually used were based on our previous experiment [31,32,33]. The same amount of saline was injected i.p. into the rats in Groups A and C. RAMTs were conducted with Group AK, Group AP, and Group APK 12 h after each anaesthesia application (simultaneously for Groups C and A), and data were recorded. According to previous studies, cognitive recovery from propofol after general anaesthesia requires 1–3 h, and reattaining the pre-anaesthetic condition requires 6 h [34,35]. It has also been determined that Y maze behaviours of rats are affected up to the 9th hour following isoflurane anaesthesia [36]. In light of this information, we decided to conduct the RAMTs 12 h after anaesthesia to prevent post-anaesthesia effects from affecting our results. Then, 24 h after final anesthesia administration, all rats were euthanized under ketamine anaesthesia, and their brain tissues were harvested. The brain tissue samples were analysed biochemically, and the prefrontal cortexes tissues were examined histopathologically.

### 2.2. Biochemical Assessment

Biochemical analyses were performed in accordance with the methods provided in our previous publications [37,38]. Brain tissue (right hemisphere) catalase (CAT) and PON-1 activity and TBARS levels were evaluated [39]. Catalase activity was measured following Aebi H’s method [40], and PON-1 enzyme activity was measured using Brites et al.’s method [41]. The protein contents of the samples were evaluated using the Lowry O technique, with BSA serving as the reference protein [42].

### 2.3. Histopathological Evaluation

The brain tissues of the experimental groups were fixed in 10% neutral formalin solution for light microscopic examinations. Paraffin blocks were obtained through routine histological follow-up procedures, and 4 µm-thick sections of the paraffin blocks were placed on milled and polylysine slides with a microtome (Leica SM 2000, Wetzlar, Germany) for haematoxylin-eosin. Images of the prepared sections were obtained using a Leica DM 4000B (Germany) computer-aided light microscope and evaluated using Leica LAS V4.9 software.

### 2.4. Immunohistochemical Analysis

For the immunohistochemical analysis, 4 μm-thick sections were extracted from the paraffin blocks of brain tissues, placed on polylysine slides, and passed through a decreasing series of alcohol. Subsequently, the sections were passed through distilled water, and a heat-induced antigen-retrieval procedure was performed with citrate buffer (Thermo, Cheshire, UK; pH 6.0). The tissues were cooled at room temperature for 20 min and then rinsed with distilled water. Following this, the tissues were washed with phosphate buffer saline (PBS; Thermo, AP-9009-10, UK; pH 7.4) and left active in 3% hydrogen peroxide (Thermo, TA-125-HP, UK). The slides were then washed with PBS, and the immunohistochemical method was continued using the Ultra Vision Detection System Large Volume Anti-Polyvalent, HRP (RTU) (Thermo, TP-125-HP, UK) kit. Nonspecific binding was prevented by applying an UltraV block, and the primary antibody stage was started without washing the tissues. Sections with appropriate dilutions of p53 (Elabscience, Houston, TX, USA; E-AB-70040; 1:100), Bcl2 (Elabscience, E-AB-60012; 1:100), Bax (Elabscience, E-AB-33819; 1:200), caspase-3 (1:100), and GFAP (Elabscience, E-AB-70040; 1:500) primary antibodies were incubated at +4 °C overnight. Following this, the slides were washed with PBS. A secondary antibody with biotin was then applied, and the slides were washed with PBS again. The tissues were left active in the streptavidin peroxidase enzyme complex and then washed with PBS. A visible immune reaction was induced with chromogen DAB (Thermo, TA-125-HD, UK) containing diaminobenzedine (DAB) substrate. Mayer’s haematoxylin (Thermo, TA-125-MH, UK) was used as the counterstain. Immunohistochemical analyses of sections covered with enthallan after 20 min in xylol were visualized under a Leica DM 4000B (Germany) computer-aided light microscope and Leica LAS V4.9 (Germany) software. Immunohistochemical analyses of the p53, Bcl2, Bax, Caspase-3, and GFAP primary antibodies in the prefrontal cortexes of the experimental groups were performed by measuring the percentages of immunopositive areas (magnification, ×400) using ImageJ software (Java-based software program, Version 1.53a, National Institutes of Health and the Laboratory for Optical and Computational Instrumentation (LOCI, University of Wisconsin)).

### 2.5. Statistical Analysis

For the statistical analysis, the Statistical Package for Social Sciences 20.0 (SPSS 20.0, Chicago, IL, USA) was used. The Shapiro–Wilk test was used for comparisons of the variable groups. Analysis of variance (ANOVA) and Bonferroni post-hoc tests were performed to assess the results, with a two-way repeated measures ANOVA for within-group RAMT comparison. All data were expressed as mean ± standard deviation (SD) values. A value of *p* < 0.05 was considered statistically significant.

## 3. Results

None of the 30 rats used in our study died. All 30 rats were subjected to histopathological evaluations and biochemical assessments. The RAMT entries and exits were initially similar in number across all groups; they decreased significantly in Groups A, AK, AP, and APK, as compared to Group C, after AD was established in the 2nd week. Compared to Group A, the RAMT entries and exits increased significantly in Groups AK and APK after the first anaesthesia and in Group AP after the second anaesthesia (Table 1). 

When the within-group differences were investigated over time based on the control values, the following results were obtained: The mean RAMT entry and exit values of the rats in Group C were similar. The mean RAMT entry and exit values of Group A decreased significantly compared to the baseline values at all measurement times, except the 1st week. The mean RAMT entry and exit values of Groups AK and APK decreased significantly compared to the initial values at all measurement times following the first anaesthesia application and before the 2nd week. The mean RAMT entry and exit values of Groups AK and APK were similar to the initial values at all measurement times after the second and third anaesthesia applications. The mean RAMT entry and exit values of Group AP decreased significantly compared to the initial value at all measurement times, except after the 1st week and after the third anaesthesia application. The mean RAMT entry and exit values of Group AP were similar to the initial values measured after the third anaesthesia application (Table 1).

PON-1 and CAT enzyme activity levels were much lower in Group A, while TBARS levels were significantly higher in Group A than in Group C (*p* < 0.05). The brain tissue TBARS levels were significantly lower in Groups AP and APK than in Group A, whereas CAT enzyme activity was significantly higher (*p* < 0.05). PON-1 activity was significantly higher in Groups AK, AP, and APK than in Group A (*p* < 0.05) (Table 2).

Neuron structures with normal histological appearances were observed in the tissue sections obtained from Group C, and no histopathological changes were observed in the neurons and glial cells. Degenerative neuron structures, granulovacuolar degeneration, and pycnotic nuclei were found in Group A. The histopathological changes partially decreased in Group AP and AK. While continued in some areas, these degenerative changes decreased more in the Group APK (Figure 1). When the capillaries were evaluated, significant perivascular edema was observed in Group A compared to Group C, and this edema was observed to partially decrease in the anesthetized groups.

The results of the immunohistochemical analyses are shown in Table 3 and Figure 2. According to the results, Bax immunopositivity was evaluated across the groups and was found to increase following STZ treatment (*p* < 0.001). The percentage of Bcl2 immunopositivity was significantly higher in Group C than in Group A (*p* < 0.001). Further, Bcl2 immunopositivity was statistically significantly elevated in Groups AK and AP compared to Group A (*p* = 0.001 and *p* < 0.001, respectively). When Groups APK and A were compared, the elevation in Bcl2 immunoreactivity observed in Group APK was statistically significant (*p* < 0.001). Caspase-3 immunopositivity was also found to increase with STZ treatment (*p* < 0.001). A significant decrease was noted in Group APK compared to Group A (*p* = 0.007). Neither ketamine nor propofol alone affected the STZ-induced increase in caspase-3 expression. Immunopositive areas belonging to the p53 antibody statistically significantly increased in the STZ treatment groups (*p* < 0.001). GFAP expression increased following STZ treatment (*p* < 0.001), and anaesthesia treatment did not affect the STZ-induced increment in GFAP expression (Table 3, Figure 2).

## 4. Discussion

Ketamine and propofol are the most commonly used intravenous anesthetic agents in all age groups today. As postoperative cognitive dysfunction is more likely to be observed in Alzheimer’s patients, it is important to investigate the effects of ketamine and propofol [4]. Therefore, the present study was conducted to evaluate the effects of repetitive anaesthesia, ketamine, and propofol administration on the cognitive functions of rats with AD. Based on this study, it was discovered that the repetitive use of anesthesia has a beneficial impact on cognitive functions. Additionally, it was observed that the administration of ketamine has a particularly strong neuroprotective effect.

Propofol can have both neurotoxic and neuroprotective effects at various doses [19,43,44,45,46,47]. Propofol’s neuroprotective effects in rodent models of traumatic brain injury are linked to antioxidant properties and the enhancement of GABA_A_ receptors, which regulate synaptic transmission inhibition and the inhibition of glutamate release [47,48]. It also causes neurotoxic effects in human stem cell-derived neurons [49] and the developing brain through a number of mechanisms, including the disruption of the blood–brain barrier [50]. Although the effects of propofol on learning and memory have been investigated in many studies, they remain controversial [21,22,23,24,25]. In an experimental study on the effects of propofol on rats’ cognitive functions, the Morris water maze (MWM) test was used to analyse cognitive function. The rats were administered a total of 30 mg/kg of propofol, and their cognitive functions were evaluated 1 h and 24 h after anaesthesia. In rats treated with propofol, escape latency was longer, and target quadrant retention time was shortened after drug administration compared to the pretreatment period. Additionally, no improvement in cognitive functions was observed after 24 h, indicating that the effect of propofol anaesthesia on the rats’ cognitive functions lasted for a long time [23]. However, in another study where chronic treatment with propofol was investigated in aged wild-type (WT) and AD Tg mice, once-weekly propofol treatment shortened the escape latency of aged WT mice during the MWM tests applied 4 weeks and 8 weeks after treatment, as compared to saline treatment. These results indicated that weekly treatment with propofol specifically improved the spatial memory ability of the aged mice. Regarding the AD Tg mice, the time to reach the platform shortened after the 12th week for the propofol-administered group, and a statistically significant difference was observed after the 16th week for the saline-administered group. Further, it was reported that a propofol treatment dose of 50 mg/kg per week could improve the cognitive functions of AD Tg mice [21].

NMDAR activation has been shown to play a role in AD-related synaptic dysfunction. The fact that glutamate is the main excitatory neurotransmitter in the brain regions affected in AD cases explains the impairment in glutamate neurotransmission that occurs with this disease [1]. Ketamine is an NMDAR antagonist. NMDAR antagonists have been reported to have neuroprotective effects [51]. Evidence from recent animal studies and human epidemiological surveys has shown that chronic ketamine exposure can cause lasting cognitive impairment in animals and humans [52,53]. For instance, in a study involving rats, ketamine and saline were administered for 4 weeks. MWM tests revealed that escape latency was reduced day by day in the saline treatment group, whereas this duration was prolonged from the 4th day, rather than being shortened, in the ketamine group. Despite abstinence from ketamine for 6 weeks, mice in the ketamine group still had long escape latencies and poor performance in a probe test. According to these findings, chronic ketamine exposure may prolong spatial learning [54]. In the current study, we performed RAMT to evaluate the cognitive function of an STZ-induced AD model following repeated ketamine and propofol anaesthesia. In our study, the RAMT entry–exit numbers were initially similar across all groups, but they decreased significantly from the 2nd week onwards in Groups A, AK, AP, and APK compared to Group C, after AD was established. Compared to Group A, the RAMT entry and exit numbers increased significantly after the first anaesthesia in Groups AK and APK, which were the ketamine-administered groups, and after the second anaesthesia application in Group AP, which involved only propofol. Zheng et al. [23] did not observe any improvement in cognitive function after administering a single dose of propofol, whereas the RAMT entries and exits in our study started to increase after the second dose of propofol. Shao et al. reported that chronic propofol use improves cognitive functions in aged WT and AD Tg mice [21]; this result is supported by the data obtained in our study. Further, it has been stated that GABA decreases in AD patients, depending on age, and that this decrease may be related to decreases in protein and mRNA levels in GABA receptor subunits [18]. These findings suggest the strategic potential of increased GABA neurotransmission in treating AD. Based on these results, we believe that repeated administration was the reason for propofol’s positive effects on the cognitive functions of rats with AD in the present study. Huang et al. reported that chronic ketamine use causes spatial learning dysfunction [54]. However, the literature also indicates that ketamine has neuroprotective effects in regulating long-term potentiation and depression while supporting cognitive function, as well as therapeutic effects on major depression and AD [1,16,51]. In our study, the increased numbers of RAMT inputs and outputs after the first ketamine administration indicate that glutamate is the main excitatory neurotransmitter in the brain regions affected by AD, highlighting the role of deteriorating glutamate neurotransmission in this disease.

In addition to NMDA-mediated excitotoxicity, oxidant stress and the suppression of cholinergic signaling are possible mechanisms for postoperative cognitive impairment or neurodegeneration. According to several studies, anaesthesia-induced tau hyperphosphorylation is associated with POCD [22,23,44,55]. In mice and SH-SY5Y neuronal cells without hypothermia, a single dose of propofol has been shown to directly increase tau hyperphosphorylation at different epitopes in vitro and in vivo by inhibiting several tau-related kinases [44]. The effects of propofol infusion (for two hours) with hypothermia or without hypothermia on different tau-related kinases, such as Akt/GSK3β (glycogen synthase kinase-3β), MAPK pathways (p44/42 mitogen-activated protein kinase), or PP2A (protein phosphatase 2A), have been determined in a region-specific manner. Consequently, propofol has been shown to increase tau phosphorylation under both normothermic and hypothermic conditions, with temperature control resulting in the partial reduction of tau hyperphosphorylation [22]. In addition, immunohistochemistry results have revealed elevated p-Tau levels in rat hippocampi 1 h and 24 h after propofol administration. Increased levels of GSK3β, tau protein (p-tau), cyclin D1, and cleaved caspase 3 (c-Caspase 3), and their mRNA expressions, were also found in propofol-treated groups compared to a control group. It was stated that propofol causes the hyperphosphorylation of tau protein and thus induces the cell cycle re-entry of mature neural cells, leading to cell apoptosis and, ultimately, impaired cognitive function [23].

It has also been stated that GABA decreases in AD patients, depending on age, and that this decrease may be related to decreases in protein levels in GABA receptor subunits and mRNA. These findings suggest that increasing GABA neurotransmission has strategic potential in treating AD [18,56]. However, it has been shown that GM1 ganglioside (GM1) binds to Aβ, inducing Aβ fibrillogenesis, and that propofol indirectly inhibits Aβ accumulation by reducing cellular GM1 expression via GABA-A receptors [57].

Aβ is a major pathogenic molecule in AD, formed by the misprocessing of APP. Monomers and oligomers of Aβ impair glutamate uptake mechanisms, resulting in the enhanced activation of extrasynaptic GluN2B-containing NMDARs [58]. Aβ oligomeric strains have been shown to induce Ca^2+^ permeability in cultured cortical neurons through the activation of GluN2B-containing NMDARs. Due to this, memantine, a noncompetitive open channel blocker of NMDARs, is mostly prescribed as memory-preserving therapy for patients with moderate or advanced AD [1].

Another view in the literature is that exposure to ketamine decreases the expression of the GABA-producing enzyme GAD67 in parvalbumin interneurons [59]. A decline in GABA levels is considered to be caused by a decrease in the quantity and quality of GABAergic interneurons. It has been reported that ketamine impairs memory mechanisms and cognitive functions by decreasing the prefrontal cortex and hippocampus gamma oscillator frequencies [54,60]. In our study, histopathological evaluations of the prefrontal cortex brain tissue of the experimental groups showed that the degenerative changes observed in the AD group, as compared to the control group, partially returned to normal. This improvement was greater in the ketamine-administered treatment group, while the histopathological changes thought to be ketamine-induced decreased in the ketamine + propofol group.

It has been reported that GFAP labeling differs across brain regions in AD patients’ brain tissues. GFAP has been found in 80% of hippocampus astrocytes [61]. In the current study, comparing the proportions of GFAP immunopositive areas between groups revealed that GFAP expression increased following STZ treatment, playing an important function in neuroinflammation associated with astrocyte activation. Anaesthesia treatment did not affect the STZ-induced increment in GFAP expression.

Caspases are important apoptosis mediators. Caspase-3 activation, caspase-3-cleaved b-actin, and caspase-cleaved APP have been discovered in the brains of AD patients [62]. Zheng et al. showed that the level and expression of cleaved caspase 3 (c-Caspase 3) increased in rats treated with propofol [23]. However, other studies have demonstrated that propofol lowers the caspase-3 activation and Aβ oligomerization caused by the anaesthetic isoflurane and can improve cognitive function in people [19,20]. The immunohistochemical analyses in our study revealed a remarkable decrease in the immunoexpression of Caspase-3 in Group APK, which was observed to be ketamine-induced. In addition, ketamine and propofol were effective at inducing antiapoptotic Bcl-2 immunoexpression, which prevents apoptosis in neurons, whereas the combined administration of ketamine and propofol had no effect on the STZ-induced increase in Bax expression.

P53 plays a considerable role in the development of neurodegenerative diseases and apoptosis. Increased p53 levels have been found in the brain tissues of patients with AD and in transgenic mice [63,64,65]. In our study, p53 immunoexpression increased following STZ treatment, but there was no difference between the treatment groups.

It has been stated that oxidative stress has an important effect on the development of AD. Oxidative stress causes the accumulation of Aβ plaque, in turn causing the formation of free radicals and oxidative stress [66]. The body’s defence mechanism against oxidative stress is formed by antioxidants, with CAT being one such antioxidant enzyme. In our study, we found that brain tissue CAT enzyme activity was lower in Group A than in Group C, whereas CAT enzyme activity was significantly higher in Groups AP and APK than in Group A.

The decrease in membrane phospholipids resulting from lipoperoxidation has also been shown to be a major cause of neurodegenerative diseases [67]. Increased lipid peroxidation products occur in sensitive cortical regions in AD brains. Circulating TBARSs provide information about the level of lipid peroxidation in the body that cannot be balanced by other tissue-derived antioxidants [67].

PON-1 is an enzyme that inhibits the buildup of lipid peroxides in low-density lipoproteins [68]. Oxidative stress and PON-1 status may play essential roles in several neurodegenerative disorders, including AD [69,70,71]. In our study, TBARS levels were statistically significantly lower in the propofol-administered groups. Brain tissue PON-1 activity was significantly higher in Groups AK, AP, and APK compared to Group A. Therefore, we determined that ketamine and propofol are both effective at evaluating PON-1 activity. Biochemical analyses also showed that both ketamine and propofol were effective in addressing the AD induced in the rats.

Lastly, the connection between anesthesia and Alzheimer’s disease is not well understood. The neuroprotective effect of ketamine administration was found to be more pronounced in the present study. In anaesthesia practice, the number of Alzheimer’s patients is continuously increasing. Propofol and ketamine are frequently employed agents in the practice of anaesthesia. It will be beneficial to establish a connection between these results and clinical practice by conducting large-scale, multicenter observational studies in the future.

The study has some limitations. First limitation: a sample size/power analysis could not be performed due to the restriction imposed by the animal research committee on the number of animals allowed. Consequently, the number of rats in each group was determined based on the committee’s authorization. Second limitation: the cumulative effect of repeated intraperitoneal ketamine (anesthetics) administration has not been evaluated. 

## 5. Conclusions

It was determined that repeated anesthesia application positively affected cognitive functions evaluated by RAMT in relation to the number of repetitions. Histopathological and biochemical indicators also show that repeated anesthesia positively affects the results in aged rats with STZ-induced AD. It was determined that the neuroprotective effect of ketamine administration was more pronounced because both the RAMT effect and the histopathological effects of ketamine were more pronounced. We recommend conducting large-scale observational studies in the future to link these findings with clinical practice.

## Figures and Tables

**Figure 1 medicina-60-01266-f001:**
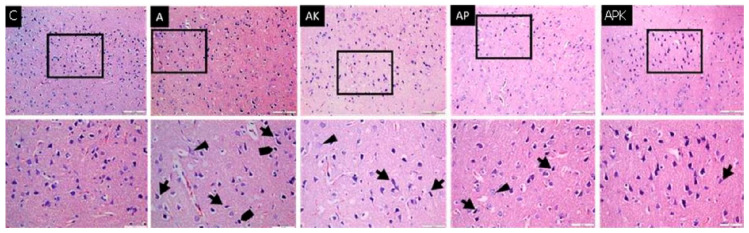
Histological images of the experimental groups; degenerative neurons (
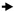
), granulovacuolar degeneration (►), and pycnotic nuclei (

), which increased in Group A and decreased in Group APK, can be observed (haematoxylin-eosin, ×200 and ×400).

**Figure 2 medicina-60-01266-f002:**
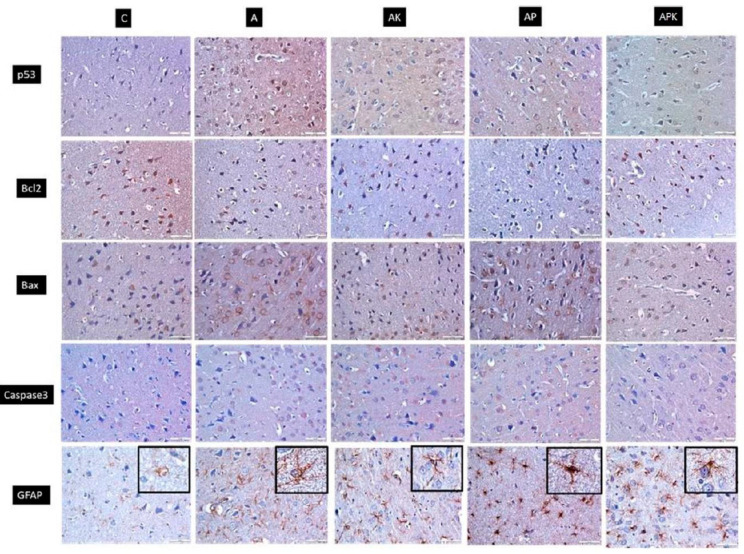
Immunohistochemical images of all groups and antibodies (DAB-haematoxylin ×400).

**Table 1 medicina-60-01266-t001:** RAMT input–output results of the rats (mean ± SD).

Rat/Input–Output	Group C (*n* = 6)	Group A (*n* = 6)	Group AK (*n* = 6)	Group AP (*n* = 6)	Group APK (*n* = 6)	*p* **
Initial	7.67 ± 1.50	8.33 ± 0.52	7.83 ± 0.41	8.00 ± 0.63	7.80 ± 0.82	0.653
Week 1	7.07 ± 0.82	7.17 ± 0.75	6.67 ± 0.92	6.50 ± 0.55	6.67 ± 0.53	0.526
Week 2	6.50 ± 1.05	5.50 ± 0.55 *, ?	5.00 ± 0.89 *, ?	5.33 ± 0.52 *, ?	5.03 ± 0.63 *, ?	0.012
Week 3	7.00 ± 1.26	4.23 ± 1.17 *, ?	3.33 ± 0.52 *, ?	4.00 ± 0.63 *, ?	4.33 ± 0.82 *, ?	<0.0001
Week 4	6.80 ± 0.89	3.83 ± 1.32 *, ?	3.17 ± 0.41 *, ?	3.33 ± 0.52 *, ?	3.67 ± 0.48 *, ?	<0.0001
After 1st anaesthesia	6.67 ± 0.52	3.50 ± 0.84 *, ?	5.50 ± 1.05 +, ?	3.83 ± 0.89 *, ?	5.00 ± 0.63 *, +, ?	<0.0001
After 2nd anaesthesia	7.33 ± 1.32	2.83 ± 0.98 *, ?	7.17 ± 1.60 +	5.17 ± 0.67 *, +, ?	5.67 ± 0.82 *, +	<0.0001
After 3rd anaesthesia	7.00 ± 0.86	3.17 ± 1.04 *, ?	8.00 ± 1.55 +	5.67 ± 0.52 +, &	6.62 ± 0.82 +	<0.0001

*p* **: significance level with ANOVA test *p* < 0.05; * *p* < 0.05: compared to Group C; + *p* < 0.05: compared to Group A; and *p* < 0.05: compared to Group AK; ? *p* < 0.05: compared to the initial value.

**Table 2 medicina-60-01266-t002:** TBARS and PON-1 and CAT enzyme activity levels in rat brain tissues (mean ± SD).

Parameters	Group C (*n* = 6)	Group A (*n* = 6)	Group AK (*n* = 6)	Group AP (*n* = 6)	Group APK (*n* = 6)	*p* **
TBARS (nmol/mg protein)	0.49 ± 0.13	1.01 ± 0.13 *	0.74 ± 0.13	0.62 ± 0.21 +	0.64 ± 0.24 +	0.002
PON-1 (IU/mg protein)	98.57 ± 30.56	47.65 ± 10.46 *	92.20 ± 51.80 +	89.03 ± 15.60 +	98.57 ± 47.39 +	0.011
CAT (IU/mg protein)	82.98 ± 31.23	51.30 ± 6.57 *	62.28 ± 11.96	73.25 ± 11.53 +	73.22 ± 19.61 +	0.025

*p* **: significance level with ANOVA test *p* < 0.05; * *p* < 0.05: compared to Group C; + *p* < 0.05: compared to Group A.

**Table 3 medicina-60-01266-t003:** Comparison of the staining quantification of apoptosis-related proteins between groups (mean ± SD).

Protein	Group C (*n* = 6)	Group A (*n* = 6)	Group AK (*n* = 6)	Group AP (*n* = 6)	Group APK (*n* = 6)	*p* **
Bax	1.56 ± 0.25	10.21 ± 0.79 *	9.85 ± 0.56 *	11.64 ± 0.66 *	9.73 ± 0.85 *	<0.0001
Bcl-2	2.60 ± 0.44	0.07 ± 0.05 *	1.38 ± 0.38 +	2.08 ± 0.33 +	1.43 ± 0.30 +	<0.0001
Caspase-3	0.92 ± 0.09	14.91 ± 0.55 *	12.18 ± 0.69 *	14.50 ± 0.79 *	9.36 ± 0.65 *, +	<0.0001
p53	2.76 ± 0.31	11.04 ± 1.23 *	7.96 ± 0.64 *	7.75 ± 0.65 *	7.36 ± 0.50 *	<0.0001
GFAP	1.93 ± 0.29	11.63 ± 1.14 *	7.69 ± 0.93 *	10.64 ± 0.96 *	8.29 ± 0.66 *	<0.0001

*p* **: significance level with ANOVA test *p* < 0.05; * *p* < 0.05: compared to Group C; + *p* < 0.05: compared to Group A.

## Data Availability

All the data generated and analyzed in this study are included in this published article.

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
