# Peer review of "Evaluation of the Effects of Repetitive Anaesthesia Administration on the Brain Tissues and Cognitive Functions of Rats with Experimental Alzheimer’s Disease†"

_medicina, 2024, doi:10.3390/medicina60081266_

Round 1

Reviewer 1 Report

Comments and Suggestions for Authors

This manuscript describes the effects of repetitive ketamine, propofol or combination ketamine & propofol on cognitive, biochemical and histopathological outcomes in a streptozocin- induced model of Alzheimer's disease. This study addresses an interesting topic and has novel findings, though it has several limitations which need to be addressed and/or more clearly articulated.

Major comments:

11.  The abstract provides sufficient information about the study design and experimental findings, though the rationale for the study is missing here. This could be improved by briefly mentioning why ketamine and propofol were chosen for investigation in the context of AD.

22.  Likewise, the concluding remarks of the abstract mention correlation of the findings with clinical procedures. This is the first time that a clinical scenario is mentioned. Introducing the clinical problem earlier will help to contextualise the research.

33. The introduction provides a comprehensive summary of the neuropathology of AD, and the relevant anaesthetics (ketamine and propofol) along with their receptor-mediated activities. What is missing is the rationale for the study – why would propofol and ketamine have effects specifically in Alzheimer's? Is there a clinical question authors are trying to answer here? A clear statement of the aims and hypotheses is needed.

44. The introduction covers a lot of ground on molecular and mechanistic aspects of tau and amyloid-beta (Aβ) pathologies and the receptor-mediated actions of ketamine and propofol. It would be beneficial to consolidate this and provide more relevant background on the outcome measures used in this study. For example: Mentioning that learning and memory deficits are notable features of AD, as well as the role of inflammation, oxidative stress, and apoptosis in cognitive decline associated with AD.

55. Six rats were used per group. How was this number reached, was it based on a power calculation? Perhaps the authors might discuss the need for greater experimental sizes and a broader panel of cognitive assessments.

66. All rats had ketamine prior to STZ – do the authors think that this dose of ketamine could have had any effect on later results?

77. Were groups A and C administered vehicle injections when the treatment groups had i.p anaesthesia injections?

88. Given that all rats were euthanised under ketamine anaesthesia 24 hours after RAMT, are the authors aware of any potential confounders for the ex vivo measures, perhaps due to the cumulative effect of ketamine administration for Groups AK and APK?

99. It is unclear which tests were done in blood, and whether this was whole blood, serum or plasma. More information is needed on this.

10. The prefrontal cortex was utilised for immunohistochemical analyses of p53, Bcl2, Bax, Caspase-3, and GFAP. Authors should state their rationale for this. Earlier in methods the authors say hippocampi were used for histopathology, though this does not appear to be the case. Given the crucial role of the hippocampus for spatial memory and navigation this structure would be important to examine as it may directly influence RAMT performance.

11. The results text is presented clearly. However, the figures could be improved for clarity: A close-up representative image of the degenerative neurons (vs. a healthy neuron) would help clarify the outcome measure. While the arrows illustrate differences in number, the distinction between staining is difficult to confirm at this resolution. Figure 2 - the representative images are difficult to confirm group differences described in the results text.

12.  Table 3 shows counts of image intensity – this is discordant with the description in the methods of measuring percentage area.

13.  It would really help the reader to understand context if the authors would use the first paragraph of the discussion to remind the reader what the aim of the study was, and whether the results supported their hypotheses.

  1. Some of the information provided requires clarification regarding its relevance to the research outcomes. Example on Page 9:“Based on these results, it can be stated that propofol causes the hyperphosphorylation of tau protein and thus induces the cell cycle re-entry of mature neural cells, leading to cell apoptosis and ultimately to impaired cognitive function”

15.  A lot of new material is introduced in the discussion that may be more beneficial in the introduction, if not removed entirely. A more focused discussion is needed in which the authors place the results of this study and the outcomes assessed in the context of the literature. At present this is more an afterthought at the end, and not the main focus of the discussion.

  1. On page 10, it would be helpful to know whether previous studies of ketamine and propofol have shown similar or contrasting effects on p53, Bcl2, Bax, Caspase-3, and GFAP immunoexpression and TBARS levels. This would provide insight into the potential modulation of inflammatory/oxidative and neurodegenerative pathways with anaesthetics like propofol and ketamine.
  2. If possible, please provide a citation (page 10): “The decrease in membrane phospholipids resulting from lipoperoxidation has also been shown to be a major cause of neurodegenerative diseases. Increased lipid peroxidation products occur in sensitive cortical regions in AD brains.”

18.  The discussion would be improved with a study limitations section and suggestions for further studies to advance the research findings presented. Expand on the potential clinical implications of your findings. Discuss how these results might inform the use of anaesthetics in patients with or at risk for AD and what future research should focus on to bridge the gap between preclinical and clinical settings.

19.  The concluding paragraph notes that histological and biochemical findings corroborate the RAMT results, showing that anaesthesia has benefits in line with the number of repetitions. This needs to be revised as all post-mortem assessments were conducted only after the maximum number of anaesthesia repeats.

Minor comments:

-  Vague sentence requires rewording: “Although degenerative changes decreased in Group APK, too, the changes continued in some areas”.

-        There was no propofol or ketamine administered to the control or alzhimer’s control groups, so no way to determine the effects of these alone. This should be stated as a limitation.

-       How was brain tissue used for enzyme assays, histology and immunohistochemistry? A clearer statement (e.g. “one hemisphere was used for…”) is needed.

-    Section of biochemical assessment contains a repeated phrase about the radial arm maze – to remove.

-        Citrate buffer antigen retrieval is not used to remove alcohol.

-    A clearer statement of timeline is needed – were rats euthanised 24 hours after the final anaesthetic administration?

-       Definition of single asterisk missing from Table 1 legend.

-      It would help the reader to see the enzyme results in their own section, rather than at the end of the RAMT results.

-        Both figures 1 and 2 need information about the size of the scale bar.

Author Response

Dear Editor,

We thank the reviewers for their valuable opinions and contributions to our study. We carefully evaluated the valuable reviewer opinions and made the necessary changes to our manuscript according to these suggestions. We indicate the changes made below.

Reviewer

This manuscript describes the effects of repetitive ketamine, propofol or combination ketamine & propofol on cognitive, biochemical and histopathological outcomes in a streptozocin- induced model of Alzheimer's disease. This study addresses an interesting topic and has novel findings, though it has several limitations which need to be addressed and/or more clearly articulated.

Major comments:

  1. The abstract provides sufficient information about the study design and experimental findings, though the rationale for the study is missing here. This could be improved by briefly mentioning why ketamine and propofol were chosen for investigation in the context of AD.
  2. Likewise, the concluding remarks of the abstract mention correlation of the findings with clinical procedures. This is the first time that a clinical scenario is mentioned. Introducing the clinical problem earlier will help to contextualise the research.
  3. The introduction provides a comprehensive summary of the neuropathology of AD, and the relevant anaesthetics (ketamine and propofol) along with their receptor-mediated activities. What is missing is the rationale for the study – why would propofol and ketamine have effects specifically in Alzheimer's? Is there a clinical question authors are trying to answer here? A clear statement of the aims and hypotheses is needed.

Response: 1, 2 and 3: According to your valuable suggestions, we incorporated sentences into the introduction to more effectively establish the rationale behind the study. Unfortunately, the abstract of the study is restricted to 300 words, as stipulated in the Guide for Authors and by Reviewer #1. We are unable to include any additional sentences into the abstract.

  1. The introduction covers a lot of ground on molecular and mechanistic aspects of tau and amyloid-beta (Aβ) pathologies and the receptor-mediated actions of ketamine and propofol. It would be beneficial to consolidate this and provide more relevant background on the outcome measures used in this study. For example: Mentioning that learning and memory deficits are notable features of AD, as well as the role of inflammation, oxidative stress, and apoptosis in cognitive decline associated with AD.

Response: In the introduction and discussion sections of our article, the occurrence of cognitive dysfunction in patients with AD has been extensively discussed in relation to inflammation, oxidative stress, and apoptosis. The article will be extended by the inclusion of additional information on this subject. The following sentences are provided in the article regarding this topic:

  • “Inflammatory processes and many other contributing factors are mentioned in the aetiology of AD. Aβ accumulation, caspase activation, and apoptosis have all been linked to AD neuropathogenesis.”
  • “Aβ is a major pathogenic molecule in AD, formed by the misprocessing of APP. Monomers and oligomers of Aβ impair glutamate uptake mechanisms, resulting in the enhanced activation of extrasynaptic GluN2B-containing NMDARs [58]. Aβ oligomeric strains have been shown to induce Ca2+ permeability in cultured cortical neurons through the activation of GluN2B-containing NMDARs. Due to this, memantine, a noncompetitive open channel blocker of NMDARs, is mostly prescribed as memory-preserving therapy for patients with moderate or advanced AD [1].”
  • “It has been stated that oxidative stress has an important effect on the development of AD. Oxidative stress causes the accumulation of Aβ plaque, in turn causing the formation of free radicals and oxidative stress [66]. The body's defence mechanism against oxidative stress is formed by antioxidants, with CAT being one such antioxidant enzyme.”
  • “Caspases are important apoptosis mediators. Caspase-3 activation, caspase-3-cleaved b-actin, and caspase-cleaved APP have been discovered in the brains of AD patients [62]. Zheng et al. showed that the level and expression of cleaved caspase 3 (c-Caspase 3) increased in rats treated with propofol [23]. However, other studies have demonstrated that propofol lowers the caspase-3 activation and Aβ oligomerization caused by the anaesthetic isoflurane and can improve cognitive function in people [19, 20].”

  1. Six rats were used per group. How was this number reached, was it based on a power calculation? Perhaps the authors might discuss the need for greater experimental sizes and a broader panel of cognitive assessments.

Response: The study was planned by taking the number of groups and subjects used in previous similar studies as an example. Examples of publications include:

    • Erkent FD, Isik B, Kucuk A, Ozturk L, Neselioglu S, Dogan HT, Guney S, Arslan M. Effects of recurrent sevoflurane anesthesia on cognitive functions with streptozotocin induced Alzheimer disease. Bratisl Lek Listy. 2019;120(12):887-893. doi: 10.4149/BLL_2019_149. PMID: 31855046.
    • Emik U, Unal Y, Arslan M, Demirel CB. The effects of memantine on recovery, cognitive functions, and pain after propofol anesthesia. Braz J Anesthesiol. 2016 Sep-Oct;66(5):485-91. doi: 10.1016/j.bjane.2015.03.002. Epub 2016 Jan 20. PMID: 27591462.

  1. All rats had ketamine prior to STZ – do the authors think that this dose of ketamine could have had any effect on later results?

Response: The rat model used in the study was designed based on the models outlined in earlier studies. Ketamine was administered to all subjects in these experiments and was shown to have no impact on the outcomes. These investigations;

    • Arras M, Autenried P, Rettich A, Spaeni D, Rülicke T. Optimization of intraperitoneal injection anesthesia in mice: drugs, dosages, adverse effects, and anesthesia depth. Comp Med. 2001 Oct;51(5):443-56.
    • Erkent FD, Isik B, Kucuk A, Ozturk L, Neselioglu S, Dogan HT, Guney S, Arslan M. Effects of recurrent sevoflurane anesthesia on cognitive functions with streptozotocin induced Alzheimer disease. Bratisl Lek Listy. 2019;120(12):887-893. doi: 10.4149/BLL_2019_149. PMID: 31855046.

  1. Were groups A and C administered vehicle injections when the treatment groups had i.p anaesthesia injections?

Response: Editing

  1. Given that all rats were euthanised under ketamine anaesthesia 24 hours after RAMT, are the authors aware of any potential confounders for the ex vivomeasures, perhaps due to the cumulative effect of ketamine administration for Groups AK and APK?

Response: Ketamine anesthesia was used for euthanasia in all groups. The cumulative effect of ketamine was not examined. The fact that the cumulative effect of ketamine was not examined is stated in the limitations section.

  1. It is unclear which tests were done in blood, and whether this was whole blood, serum or plasma. More information is needed on this.

Response: Biochemical and histological evaluations were performed on brain tissue. ''The brain tissue samples were analysed biochemically, and the prefrontal cortexes tissues were examined histopathologically'' was edited in the text. Editing.

  1. The prefrontal cortex was utilised for immunohistochemical analyses of p53, Bcl2, Bax, Caspase-3, and GFAP. Authors should state their rationale for this. Earlier in methods the authors say hippocampi were used for histopathology, though this does not appear to be the case. Given the crucial role of the hippocampus for spatial memory and navigation this structure would be important to examine as it may directly influence RAMT performance.

Response: Histopathological examinations were conducted in the prefrontal cortex. The article includes the immunohistochemistry analysis part where this information can be found.

  1. The results text is presented clearly. However, the figures could be improved for clarity: A close-up representative image of the degenerative neurons (vs. a healthy neuron) would help clarify the outcome measure. While the arrows illustrate differences in number, the distinction between staining is difficult to confirm at this resolution. Figure 2 - the representative images are difficult to confirm group differences described in the results text.

Response: First of all, thank you very much for your careful review and suggestions. Since there is a lot of group and histological data in the study, the pictures were necessarily reduced and added to the manuscript. However, the resolution of the images has been updated to 600Dpi in order to increase the image quality and make the histological changes more distinguishable. In this way, the clarity of the picture will be preserved when the page is enlarged.

  1. Table 3 shows counts of image intensity – this is discordant with the description in the methods of measuring percentage area.

Response: Editing

  1. It would really help the reader to understand context if the authors would use the first paragraph of the discussion to remind the reader what the aim of the study was, and whether the results supported their hypotheses.

Response: According to your suggestion, we incorporated sentences into the first paragraph of discussion to remind the reader the aim of the study, and results.

  1. Some of the information provided requires clarification regarding its relevance to the research outcomes. Example on Page 9:“Based on these results, it can be stated that propofol causes the hyperphosphorylation of tau protein and thus induces the cell cycle re-entry of mature neural cells, leading to cell apoptosis and ultimately to impaired cognitive function”

Responese: The information sentence given was re-written.

  1. A lot of new material is introduced in the discussion that may be more beneficial in the introduction, if not removed entirely. A more focused discussion is needed in which the authors place the results of this study and the outcomes assessed in the context of the literature. At present this is more an afterthought at the end, and not the main focus of the discussion.
  1. On page 10, it would be helpful to know whether previous studies of ketamine and propofol have shown similar or contrasting effects on p53, Bcl2, Bax, Caspase-3, and GFAP immunoexpression and TBARS levels. This would provide insight into the potential modulation of inflammatory/oxidative and neurodegenerative pathways with anaesthetics like propofol and ketamine.

Response 15 and 16: After conducting an extensive search, we found no existing studies on this subject in the literature. Previous research has examined the effects of propofol and ketamine, although different biomarkers were utilized instead of the ones we employed in our study. The discussion primarily focuses on the existing studies mentioned in the literature. There are no existing studies that precisely align with our research.

  1. If possible, please provide a citation (page 10): “The decrease in membrane phospholipids resulting from lipoperoxidation has also been shown to be a major cause of neurodegenerative diseases. Increased lipid peroxidation products occur in sensitive cortical regions in AD brains.”

Response: We added the reference at the end of sentence, as you recommended.

  1. The discussion would be improved with a study limitations section and suggestions for further studies to advance the research findings presented. Expand on the potential clinical implications of your findings. Discuss how these results might inform the use of anaesthetics in patients with or at risk for AD and what future research should focus on to bridge the gap between preclinical and clinical settings.

Response: As same as your suggestion, Reviewer #2 previosly suggested this to enhance the study’s overall value. A paragraph was added into discussion, about the recommendations for future research directions.

“Lastly, the connection between anesthesia and Alzheimer's disease is not well un-derstood. The neuroprotective effect of ketamine administration was found to be more pronounced in the present study. In anaesthesia practice, the number of Alzheimer's pa-tients is continuously increasing. Propofol and ketamine are frequently employed agents in the practice of anaesthesia. It will be beneficial to establish a connection between these results and clinical practice by conducting large-scale, multicenter observational studies in the future.”

Response Added. No power analysis was conducted to determine the appropriate sample size. Each experimental group consisted of six animals, in adherence to the guidelines outlined by the ethics committee. The inclusion of six animals per group represents the maximum permissible number sanctioned by the ethics committee under the purview of the 3Rs rule (replacement, reduction, and refinement).

  1. The concluding paragraph notes that histological and biochemical findings corroborate the RAMT results, showing that anaesthesia has benefits in line with the number of repetitions. This needs to be revised as all post-mortem assessments were conducted only after the maximum number of anaesthesia repeats.

Response: The sentences in the conclusion, was re-written according to your suggestion.

Minor comments:

-  Vague sentence requires rewording: “Although degenerative changes decreased in Group APK, too, the changes continued in some areas”.

Response: This sentence in the abstract was deleted.

-        There was no propofol or ketamine administered to the control or alzhimer’s control groups, so no way to determine the effects of these alone. This should be stated as a limitation.

Response: Added

-       How was brain tissue used for enzyme assays, histology and immunohistochemistry? A clearer statement (e.g. “one hemisphere was used for…”) is needed.

Response: Editing

-    Section of biochemical assessment contains a repeated phrase about the radial arm maze – to remove.

Response: The sentence in the biochemical assessment section contains RAMT was deleted.

-        Citrate buffer antigen retrieval is not used to remove alcohol.

Response: Corrected

-    A clearer statement of timeline is needed – were rats euthanised 24 hours after the final anaesthetic administration?

Response: This sentence was re-written according to your suggestion

-       Definition of single asterisk missing from Table 1 legend.

Response: Editing

-      It would help the reader to see the enzyme results in their own section, rather than at the end of the RAMT results.

Response: The RAMT results and enzyme results were given in two different paragraphs.

-        Both figures 1 and 2 need information about the size of the scale bar.

Response: First of all, thank you very much for your careful review and suggestions. Since there is a lot of group and histological data in the study, the pictures were necessarily reduced and added to the manuscript. However, the resolution of the images has been updated to 600Dpi in order to increase the image quality and make the histological changes more distinguishable. In this way, the clarity of the picture will be preserved when the page is enlarged.

Best regards.

Reviewer 2 Report

Comments and Suggestions for Authors

The abstract is much too long, it must be limited to 300 words. Also, a background of the study must be added before the purpose.

The introduction can be shortened. For example, the paragraph that begins "Gamma oscillations, in addition to synaptic plasticity..." can be removed or moved to discussions.

Define STZ on first use in the text, not in the abstract.

Table 3 – how you quantified the immunohistochemical intensities.

Given the fact that it is an amyloidosis, can you write in detail the appearance of the vessels? Were changes observed at their level?

The conclusions must be made as a separate paragraph.

Author Response

Dear Editor,

We thank the reviewers for their valuable opinions and contributions to our study. We carefully evaluated the valuable reviewer opinions and made the necessary changes to our manuscript according to these suggestions. We indicate the changes made below.

Reviewer #1 (Blue)

The abstract is much too long, it must be limited to 300 words. Also, a background of the study must be added before the purpose.

Response:        As you stated, the word count of abstract was reduced to 300 words.

The introduction can be shortened. For example, the paragraph that begins "Gamma oscillations, in addition to synaptic plasticity..." can be removed or moved to discussions.

Response:  The sentence “Gamma oscillations, in synaptic plasticity ..” was removed.

Define STZ on first use in the text, not in the abstract.

Response:  The abbrevation of STZ was removed from the abstract, and given in the text.

Table 3 – how you quantified the immunohistochemical intensities.

Given the fact that it is an amyloidosis, can you write in detail the appearance of the vessels? Were changes observed at their level?

Response: Data shown in Table 3; As stated in the last sentence under the heading "Immunohistochemical Analysis" on page 4, it was obtained by using the ImageJ program by using the data obtained from the staining differences.

In this study, oxidant parameters were examined biochemically, proteins associated with apoptosis and GFAB, a marker of astrocytes, which is an important component of the blood-brain barrier, were examined immunohistochemically. GFAB retention values provide information about the continuity of the barrier. No marker specific for vascular endothelial cells and Amiloid-beta was used.

The conclusions must be made as a separate paragraph.

Response: The conclusions were given as a seperate paragraph.

Best regards.

Reviewer 3 Report

Comments and Suggestions for Authors

I have thoroughly reviewed the study investigating the effects of repeated anesthesia in rats with Alzheimer's disease, focusing particularly on the clinical and histopathological changes observed. First and foremost, I commend the research team for employing rigorous methodologies and addressing an interesting and comprehensive research topic.

Exploring the impact of repeated anesthesia and perioperative processes on perioperative cognitive dysfunction, especially in the elderly population, represents a pertinent area of current research. In this regard, it is noteworthy that this study contributes significantly to anesthesia practice and clinical understanding, aiming to pave the way for future research studies.

The study employs a rigorous methodology that includes appropriate controls, thorough induction of AD pathology, and comprehensive outcome measures encompassing behavior, biochemistry, and histopathology. Additionally, the use of multiple time points and repeated measures enhances the robustness of the findings, constituting significant strengths of the study.

Furthermore, the inclusion of well-designed tables and figures in the results sections helps to provide clarity, facilitating a better understanding of the study.

Several potential concerns have come to my attention, which I believe warrant consideration. 

To mitigate bias, detailed descriptions of randomization procedures and blinding protocols should be provided during evaluations. Additionally, presenting preliminary analyses or power analyses, if available, detailing the number of subjects would further enhance the methodological rigor of the study.

It would also be beneficial to provide more detailed information on the selection criteria and timing of cognitive tests used to increase methodological transparency.

Moreover, I encourage the authors to highlight more specifically the clinical implications of their findings and provide concrete recommendations for future research directions. Additionally, a more detailed discussion section on the practical applications and limitations of the study's findings in clinical settings would enhance the study's overall value.

Best regards.

Author Response

Dear Editor,

We thank the reviewers for their valuable opinions and contributions to our study. We carefully evaluated the valuable reviewer opinions and made the necessary changes to our manuscript according to these suggestions. We indicate the changes made below.

Reviewer #2 (Red)

I have thoroughly reviewed the study investigating the effects of repeated anesthesia in rats with Alzheimer's disease, focusing particularly on the clinical and histopathological changes observed. First and foremost, I commend the research team for employing rigorous methodologies and addressing an interesting and comprehensive research topic.

 Exploring the impact of repeated anesthesia and perioperative processes on perioperative cognitive dysfunction, especially in the elderly population, represents a pertinent area of current research. In this regard, it is noteworthy that this study contributes significantly to anesthesia practice and clinical understanding, aiming to pave the way for future research studies.

 The study employs a rigorous methodology that includes appropriate controls, thorough induction of AD pathology, and comprehensive outcome measures encompassing behavior, biochemistry, and histopathology. Additionally, the use of multiple time points and repeated measures enhances the robustness of the findings, constituting significant strengths of the study.

Furthermore, the inclusion of well-designed tables and figures in the results sections helps to provide clarity, facilitating a better understanding of the study.

Several potential concerns have come to my attention, which I believe warrant consideration. 

To mitigate bias, detailed descriptions of randomization procedures and blinding protocols should be provided during evaluations. Additionally, presenting preliminary analyses or power analyses, if available, detailing the number of subjects would further enhance the methodological rigor of the study.

Response: Editing. No power analysis was conducted to determine the appropriate sample size. Each experimental group consisted of six animals, in adherence to the guidelines outlined by the ethics committee. The inclusion of six animals per group represents the maximum permissible number sanctioned by the ethics committee under the purview of the 3Rs rule (replacement, reduction, and refinement).

It would also be beneficial to provide more detailed information on the selection criteria and timing of cognitive tests used to increase methodological transparency.

Response: The study was planned by taking the number of groups and subjects used in previous similar studies as an example. Examples of publications include:

  • Erkent FD, Isik B, Kucuk A, Ozturk L, Neselioglu S, Dogan HT, Guney S, Arslan M. Effects of recurrent sevoflurane anesthesia on cognitive functions with streptozotocin induced Alzheimer disease. Bratisl Lek Listy. 2019;120(12):887-893. doi: 10.4149/BLL_2019_149. PMID: 31855046.
  • Emik U, Unal Y, Arslan M, Demirel CB. The effects of memantine on recovery, cognitive functions, and pain after propofol anesthesia. Braz J Anesthesiol. 2016 Sep-Oct;66(5):485-91. doi: 10.1016/j.bjane.2015.03.002. Epub 2016 Jan 20. PMID: 27591462.
  • RAM was designed by Olton and Samuelson and is a frequently used method for the evaluation of cognitive functions in experimental animal models. For this reason, it was the preferred method to evaluate cognitive functions in our study. This information has been added to the Materials and Methods section.
  • According to previous studies, cognitive recovery from propofol after general anaesthesia takes 1–3 hours, and reattaining the pre-anaesthetic condition takes six hours. It has also been determined that Y maze behaviours of rats are affected up to the ninth hour following isoflurane anaesthesia. In light of this information, we decided to conduct the RAMTs 12 hours after anaesthesia to prevent post-anaesthesia effects from affecting our results.
    • RD, M., Intravenous anesthetics, in Miller’s anesthesia. 2010. p. 317-377.
    • Sanou, J., et al., Cognitive sequelae of propofol anaesthesia. Neuroreport, 1996. 7(6): p. 1130-2
    • Peng M, Zhang C, Dong Y, Zhang Y, Nakazawa H, Kaneki M, Zheng H, Shen Y, Marcantonio ER, Xie Z. Battery of behavioral tests in mice to study postoperative delirium. Sci Rep. 2016 Jul 20;6:29874. doi: 10.1038/srep29874. PMID: 27435513; PMCID: PMC4951688
    • Navarro KL, Huss M, Smith JC, Sharp P, Marx JO, Pacharinsak C. Mouse Anesthesia: The Art and Science. ILAR J. 2021 Dec 31;62(1-2):238-273. doi: 10.1093/ilar/ilab016. PMID: 34180990; PMCID: PMC9236661.

Moreover, I encourage the authors to highlight more specifically the clinical implications of their findings and provide concrete recommendations for future research directions. Additionally, a more detailed discussion section on the practical applications and limitations of the study's findings in clinical settings would enhance the study's overall value.

Response:  As you suggested to enhance the study's overall value, a paragraph was added into discussion, about the recommendations for future research directions.

Best regards.

Round 2

Reviewer 2 Report

Comments and Suggestions for Authors

Regarding immunohistochemistry, correct the word intensity with quantification in the table.

Regarding the vascular appearance in amyloid deposition, a microscopic description may be sufficient, without specific markers or special staining.

Author Response

Dear Editor,

We thank the reviewers for their valuable opinions and contributions to our study. We carefully evaluated the valuable reviewer opinions and made the necessary changes to our manuscript according to these suggestions. We indicate the changes made below.

Reviewer #2 (Red)

Comments: Regarding immunohistochemistry, correct the word intensity with quantification in the table.

Response: Editing

Comments: Regarding the vascular appearance in amyloid deposition, a microscopic description may be sufficient, without specific markers or special staining.

Response: Editing